# Clinical Validation of a Deep-Learning Segmentation Software in Head and Neck: An Early Analysis in a Developing Radiation Oncology Center

**DOI:** 10.3390/ijerph19159057

**Published:** 2022-07-25

**Authors:** Andrea D’Aviero, Alessia Re, Francesco Catucci, Danila Piccari, Claudio Votta, Domenico Piro, Antonio Piras, Carmela Di Dio, Martina Iezzi, Francesco Preziosi, Sebastiano Menna, Flaviovincenzo Quaranta, Althea Boschetti, Marco Marras, Francesco Miccichè, Roberto Gallus, Luca Indovina, Francesco Bussu, Vincenzo Valentini, Davide Cusumano, Gian Carlo Mattiucci

**Affiliations:** 1Radiation Oncology, Mater Olbia Hospital, 07026 Olbia, Italy; andrea.daviero@materolbia.com (A.D.); alessia.re@materolbia.com (A.R.); francesco.catucci@materolbia.com (F.C.); claudio.votta@guest.policlinicogemelli.it (C.V.); domi.piro@libero.it (D.P.); carmela.didio@materolbia.com (C.D.D.); martina.iezzi@materolbia.com (M.I.); francesco.preziosi@materolbia.com (F.P.); althea_boschetti@libero.it (A.B.); marco.marras@materolbia.com (M.M.); giancarlo.mattiucci@materolbia.com (G.C.M.); 2UOC Radioterapia Oncologica, Dipartimento di Diagnostica per Immagini, Radioterapia Oncologica ed Ematologia, Fondazione Policlinico Universitario Agostino Gemelli IRCCS, 00168 Roma, Italy; francesco.micciche@policlinicogemelli.it (F.M.); luca.indovina@policlinicogemelli.it (L.I.); vincenzo.valentini@policlinicogemelli.it (V.V.); 3UO Radioterapia Oncologica, Villa Santa Teresa, 90011 Bagheria, Italy; antoniopiras88@gmail.com; 4Medical Physics, Mater Olbia Hospital, 07026 Sassari, Italy; sebastiano.menna@materolbia.com (S.M.); flavio.quaranta@medipass.it (F.Q.); davide.cusumano@materolbia.com (D.C.); 5Otolaryngology, Mater Olbia Hospital, 07026 Sassari, Italy; roberto.gallus@materolbia.com; 6Otolaryngology, Azienda Ospedaliero Universitaria di Sassari, 07100 Sassari, Italy; fbussu@uniss.it; 7Dipartimento delle Scienze Mediche, Chirurgiche e Sperimentali, Università di Sassari, 07100 Sassari, Italy; 8Dipartimento di Scienze Radiologiche ed Ematologiche, Università Cattolica del Sacro Cuore, 00168 Roma, Italy

**Keywords:** head and neck, radiotherapy artificial intelligence, deep-learning, auto-contouring

## Abstract

Background: Organs at risk (OARs) delineation is a crucial step of radiotherapy (RT) treatment planning workflow. Time-consuming and inter-observer variability are main issues in manual OAR delineation, mainly in the head and neck (H & N) district. Deep-learning based auto-segmentation is a promising strategy to improve OARs contouring in radiotherapy departments. A comparison of deep-learning-generated auto-contours (AC) with manual contours (MC) was performed by three expert radiation oncologists from a single center. Methods: Planning computed tomography (CT) scans of patients undergoing RT treatments for H&N cancers were considered. CT scans were processed by Limbus Contour auto-segmentation software, a commercial deep-learning auto-segmentation based software to generate AC. H&N protocol was used to perform AC, with the structure set consisting of bilateral brachial plexus, brain, brainstem, bilateral cochlea, pharyngeal constrictors, eye globes, bilateral lens, mandible, optic chiasm, bilateral optic nerves, oral cavity, bilateral parotids, spinal cord, bilateral submandibular glands, lips and thyroid. Manual revision of OARs was performed according to international consensus guidelines. The AC and MC were compared using the Dice similarity coefficient (DSC) and 95% Hausdorff distance transform (DT). Results: A total of 274 contours obtained by processing CT scans were included in the analysis. The highest values of DSC were obtained for the brain (DSC 1.00), left and right eye globes and the mandible (DSC 0.98). The structures with greater MC editing were optic chiasm, optic nerves and cochleae. Conclusions: In this preliminary analysis, deep-learning auto-segmentation seems to provide acceptable H&N OAR delineations. For less accurate organs, AC could be considered a starting point for review and manual adjustment. Our results suggest that AC could become a useful time-saving tool to optimize workload and resources in RT departments.

## 1. Introduction

Head and neck (H&N) cancers represent one of the main challenges in radiation oncology, with about 9900 new diagnoses in Italy in 2020. They represent a major problem in oncology because of the need for multidisciplinary approaches and close clinical and social correlation [1,2].

Radiation therapy (RT) plays a key role in the multidisciplinary management of H&N neoplasms. Single-modality RT is considered an alternative approach to surgery in cases of early-stage disease, in some specific sub-sites of the head and neck region such as nasopharynx and oropharynx carcinoma [3]. In locally advanced disease settings, RT plays a central role in the management of H&N cancers, in combination with surgery and systemic therapies, with evaluations strongly based on patients’ age and comorbidities in the frame of a multidisciplinary and personalized approach [3,4,5].

In recent years, RT techniques have undergone significant evolutions, with new diagnostic modalities for defining target structures, as well as the introduction of intensity-modulated (IMRT) and volumetric-modulated (VMAT) treatments, leading to an optimization of targets dose coverage while also reducing doses to the organs at risk [6,7,8]. This feature is extremely relevant in the treatment of head and neck districts, where the location of the main disease and the proximity to organs at risk (OARs) with the associated risk of toxicity make the radiation treatment of H&N cancer patients most complex and time-consuming for a radiation oncology team. In H&N cancers, OAR delineation follows strict guidelines, as various studies have demonstrated that OAR and target volume contouring may be affected by significant inter-observer variability [9,10,11,12].

The diffusion of artificial intelligence (AI) is changing the workflow of RT treatment in different scenarios [13,14,15], and AI-based auto-contouring (AC) software have been developed and offered to clinicians to optimize the contouring process [16]. Deep-learning (DL) models make use of multi-layer neural networks to process large amounts of data and define the RT volumes of interest. Various experiences regarding clinical validation of AC models have already been reported in the literature, including testing AI-based delineation in different anatomical districts with single-institution workflow [17]. These trials tested AC models in multi-institutions validation groups, with the stated aim of demonstrating the validity of such systems in more complex RT treatment scenarios [18,19]. Furthermore, various studies have demonstrated that manual segmentation suffers from high inter-observers variability, and approaches to reduce such a source of uncertainty have been the subject of recent studies [20,21].

Besides reducing clinical workload, AC systems aim to reduce variability by increasing consistency in delineation among different centers that follow the same guidelines. In the context of H&N tumors, various studies have underlined the promising role of AC systems in reducing inter-observer variability and shortening the duration of the contouring process [22,23,24,25].

The impact of such solutions can assume a key role in the context of newly developing RT centers: the aim of this work was to clinically validate the use of an AI-based AC software in the H&N district during the start-up phases of an RT center, evaluating the impact of the variations performed by radiation oncologists starting from the segmentations proposed by the AI-based software.

## 2. Materials and Methods

### 2.1. Patients

Following the start of clinical activity at the Mater ART Radiation Oncology Department of Mater Olbia Hospital (Olbia, Sassari, Italy), patients affected by H&N cancer and subject to RT treatments were consecutively selected for the analysis of this study. At the time of the analysis, all the patients were of legal age and had signed an informed consent form for data collection.

A total of 12 patients treated at our center during the first months of clinical activity were analyzed. On the first day, a customized thermoplastic mask was made to ensure immobilization and allow reproducibility during the RT treatment. On the second day, a computed tomography (CT) scan was acquired using the customized immobilization system and a dedicated simulation CT (GE RT discovery, GE Healthcare, Chicago, IL, USA), following an axial acquisition protocol with 0.625 mm slice thickness, 120 kV and 260 mAs.

These patients were treated with long-course radiotherapy, with a total dose between 66 and 70 Gy delivered in 30–35 fractions.

The acquired CT images were exported from the CT workstation and processed using the auto-segmentation software of the study (Limbus Contour auto-segmentation software, version 1.0.22, AI Limbus Inc., 2076 Athol Street, Regina, SK S4T 3E5, Canada). The auto-segmentation models used in this study were described in terms of software characteristics, libraries and models in previous publications [17,18].

### 2.2. Auto-Segmentation Process

The Limbus H&N template was selected to generate the following deep-learning auto-contours (DC): bilateral brachial plexus, brain, brainstem, bilateral cochlea, pharyngeal constrictors, eye globes, bilateral lens, mandible, optic chiasm, bilateral optic nerves, oral cavity, bilateral parotids, spinal cord, bilateral submandibular glands, lips and thyroid.

The larynx was excluded from the present analysis due to substantial differences between the contour proposed by the software and the guidelines followed by our center (consisting in the division of the organs in sub-parts according to Brouwer’s definition) [9,12]. Target structures were also excluded from the analysis considering the high variability in delineation, especially for nodal targets.

The RT structure set generated by Limbus was transferred into the Varian Eclipse workstation (version 16.1, Varian Medical System, Mountain View, CA, USA) for revision and evaluation. The contours generated by the AS were manually reviewed by expert radiation oncologists (ROs) according to international consensus guidelines for H&N OAR contours [9,12,26,27]. After contour validation, target volumes were manually delineated, and treatment plans were generated.

The manual and auto-contours generated were exported as DICOM files and transferred to the MIM workstation (MAESTRO MIM, version 7.1.7, Maastricht, The Netherlands), which was used to perform the contour evaluation.

### 2.3. Evaluation Metrics

The contours generated automatically by the software were compared with those manually obtained by ROs using various similarity indices.

Yeghiazaryan et al. classified similarity evaluation metrics in three main groups: size-based methods, overlap-based methods and surface distance-based measurements [28].

To provide a general description of the contours generated, we combined an overlap-based method (Dice similarity coefficient, DSC) with a surface-based one, the 95% Hausdorff distance (95% HD).

DSC is defined by the formula
(1)$$\mathrm{DSC}\left(A,\;B\right)=\frac{2\left|A\cap B\right|}{\left|A\right|+\left|B\right|}$$
with A and B representing the automatic and manual contour volumes, respectively. This index evaluates the intersection of segmentation volumes with respect to the total area, with values ranging from 0 (no overlay) to 1 (total overlay).

The HD is a quantification of the maximum distance between the manual and automatic contour surfaces. This indicator quantifies the maximum distance between two contours by calculating the distance to the nearest point in both directions, from contour A to B and vice versa.
(2)$$\mathrm{HD}\left(A,B\right)=\max\left(h\left(A,B\right),h\left(B,A\right)\right)$$
where h(A,B) is the Euclidean distance between the voxels a and b belonging to the contours A and B, and its formulation is:(3)$$h\left(A,B\right)=\max_{a\in A}\min_{b\in B}\left|\left|a-b\right|\right|$$

The 95% HD represents the distance corresponding to the largest surface-to-surface separation among the closest 95% of surface points. Good agreement between manual and automatic contours results in 95%HD values close to zero.

## 3. Results

Patients with H&N tumors undergoing RT treatments from September 2021 to February 2022 were included in the analysis.

The study was focused on the evaluation of 23 OAR structures, resulting in a total of 276 contours analyzed. Submandibular glands were excluded from two patients as the structures were included in CTV definition.

Higher mean values of DSC were observed for the brain (1.00), eye globes (0.98) and and the mandible (0.98).

A worse overlap between automatic and manual contours was observed in left cochlea (mean DSC of 0.58), right cochlea (0.58) and optic chiasm (0.50). The remaining structures were found to have high DSC values, with mean values ranging from 0.82 to 0.97.

The analysis of 95% HD shows high distances for pharyngeal constrictors (mean value of 17.59 mm) and oral cavity (12.67 mm). Low 95% HD mean values were observed for the brain (5.35), eye globes (1.03 left, 1.13 right) and lenses (0.75 left, 0.57 right). Comparison metrics for DSC and HD are summarized in Table 1.

Figure 1 and Figure 2 show boxplot analysis for Dice and Hausdorff distance, respectively.

Large variability in terms of DSC was observed for right and left cochlea, constrictors and chiasm.

As for 95% HD values, a wide range of values was reported for parotids, constrictors and the oral cavity.

## 4. Discussion

H&N cancer management represents one of the most relevant challenges in the RT scenario. Inaccuracies in the delineation of OARs and target structures can be among the main factors limiting the feasibility and effectiveness of RT treatments.

Contour delineation is still among the few procedures of the RT workflow with prevalent manual management, which can result in a high risk in inter-observer variability, and above all, it represents one of the phases with the highest time-consuming rate of all the RT workflows [9,10,11,12].

The clinical implementation of AC systems is providing benchmark data for the comparison and validation of such systems. Various studies have tested the potential role of AC software in different districts, obtaining results that seem to suggest the possibility of developing such systems in clinical practice [17,18,19,22,23,25,29,30,31].

A new paradigm in the field of auto-segmentation has been introduced with the shift from atlas-based (ABAS) to deep-learning-based approach. A recent systematic review has analyzed current evidence in the field of H&N AC [32]. Analyzing the performance of ABAS and DL auto-segmentation approaches, better results were found in terms of metrics comparison for DL-based systems. Furthermore, the authors emphasized that the possibility to train DL models on increasingly wider datasets will result in higher future reliability of these tools.

Focusing on the AC tool analyzed in our study, Wong et al. evaluated the introduction of Limbus auto-segmentation system in different anatomical districts. The results of these studies, obtained considering volume differences and feedback from different ROs, showed that well-trained AC models are associated with positive user experience and do not require large manual editing [17,18]. The analysis performed in [17,18] did not focus on a single district, but it evaluated different structures in different anatomical districts. Even though the study of Wong et al. did not specifically take into consideration all the OARs analyzed in our study, it can be considered a benchmark for reference and comparison.

The results of this study are in line with those reported in the literature and can be considered as valuable confirmation of the reliability of this software in the H&N district. In this study, most of the structures analyzed reported DSC values higher than 0.85, with lower values in terms of 95% HD. Specifically, the structures with the highest DSC values were the brain, left and right eye globes, and the mandible, the values being 1.00 for the brain and 0.98 for the remaining structures, respectively. The 95% HD values for these structures were also good, although the values for the mandible (5.93 mm) were slightly higher than in other experiments.

Good results in terms of similarity indices were observed for bilateral brachial plexus, bilateral parotids, submandibular glands, lips, spinal cord and brainstem, with DSC higher than 0.93 and HD lower than 4.36 mm; the same volumes obtained limited 95% HD values, except for the oral cavity and left and right parotids (11.56, 9.44 and 7.50 mm, respectively): this could be the result of a more accurate manual delineation dictated by the correlation between volume and constraint dose for the evaluation of the risk of toxicity.

The structures that underwent the greatest changes after manual contour review were the optic chiasm, optic nerves and cochleae. The data for the optic nerves and chiasma are consistent with those previously reported in other studies, with DSC values ranging from 0.56 and 0.58 [18]. As reported in the mentioned studies, the reasons for this evidence may lie in the larger inter-observer variability and the difficulty in the delineation of structures such as the optic chiasm on CT images.

Regarding the comparison of cochlear volumes, a possible explanation for the variability found is the inability of AC systems to utilize CT scan window width differences despite having dedicated CT sequences: such variability led to the need for significant changes in these volumes.

The main limitation of this study is the reduced sample size, mainly due to the reduced enrollment time (6 months) and the developing nature of the RT center, which was inaugurated one month before the start of the study.

Due to the retrospective nature of this study, another criticality was the lack of a direct quantification of the time advantage obtained by using such software. However, a recent publication evaluating the same software quantified the gain in time in at least 26 min for H&N, considering that an expert observer takes 26.6 min to perform a full delineation in this district, while the software performs the same task in 0.6 min [17].

Further studies including larger cohorts of patients, supplementary information and different observers would certainly support the adoption of such AI-based systems in clinical practice.

## 5. Conclusions

The results of this study are in line with those previously published, demonstrating substantial reliability of the AS software in delineating OAR volumes in the H&N district. The added value of this research is to demonstrate the validity of these tools in real-life experience, mainly in a developing RT center. These results also showed that AS software could provide a valuable starting point for review and manual adjustment, making it a useful time-saving tool for optimizing workload and resources in the RT workflow.

The clinical implementation of such systems requires limited manual revisions from radiation oncologists, significantly supporting the contouring process in this district, which can take several hours if done manually from scratch.

## Figures and Tables

**Figure 1 ijerph-19-09057-f001:**
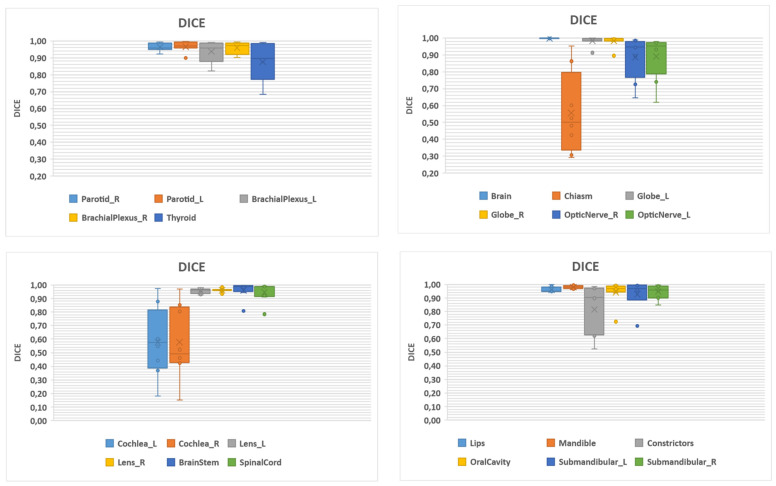
Boxplot analysis of DICE.

**Figure 2 ijerph-19-09057-f002:**
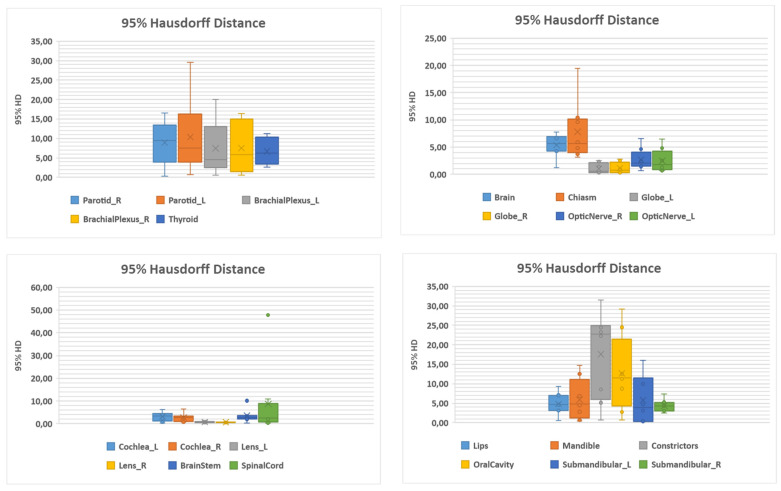
Boxplot analysis of 95% Hausdorff distance.

**Table 1 ijerph-19-09057-t001:** Values obtained quantifying the comparison between manual and automated contours in terms of DICE (DSC) and 95% Hausdorff distance (HD) for the organs object of the study.

Structure	Number	DSC		95% HD (mm)	
		Mean (SD)	Median (Range)	Mean (SD)	Median (Range)
Brachial plexus L	12	0.94 (0.06)	0.96 (0.82–0.99)	7.47 (6.71)	4.51 (0.58–20.01)
Brachial plexus R	12	0.96 (0.04)	0.97 (0.90–0.99)	7.53 (6.46)	5.84 (0.52–16.42)
Brain	12	1.00 (0.01)	1.00 (0.99–1.00)	5.35 (7.90)	5.60 (1.26–7.79)
Brainstem	12	0.96 (0.06)	0.99 (0.81–0.1)	3.46 (2.88)	2.78 (0.29–10.12)
Cochlea L	12	0.58 (0.26)	0.57 (0.18–0.98)	3.11 (1.99)	3.34 (0.29–6.24)
Cochlea R	12	0.58 (0.27)	0.49 (0.15–0.97)	2.69 (1.88)	2.78 (0.73–6.51)
Optic chiasm	12	0.56 (0.24)	0.50 (0.29–0.95)	7.79 (5.36)	5.63 (3.20–19.41)
Pharyngeal constrictors	12	0.82 (0.19)	0.90 (0.52–0.99)	17.59 (11.15)	22.71 (0.74–31.52)
Eye globe L	12	0.98 (0.03)	1.00 (0.91–1.00)	1.03 (0.93)	0.59 (0.27–2.50)
Eye globe R	12	0.98 (0.04)	1.00 (0.89–1.00)	1.13 (1.01)	0.75 (0.29–2.84
Lens L	12	0.96 (0.02)	0.96 (0.92–0.98)	0.75 (0.39)	0.58 (0.28–1.44)
Lens R	12	0.96 (0.01)	0.96 (0.93–0.98)	0.57 (0.16)	0.58 (0.27–0.74)
Lips	12	0.96 (0.02)	0.95 (0.94–1.00)	4.79 (2.83)	4.62 (0.53–9.37)
Mandible	12	0.98 (0.01)	0.98 (0.96–1.00)	5.93 (5.26)	4.75 (0.37–14.72)
Optic nerve L	12	0.89 (0.14)	0.95 (0.62–0.98)	2.67 (1.96)	2.03 (0.65–6.57)
Optic nerve R	12	0.89 (0.13)	0.95 (0.65–0.99)	2.49 (2.09)	1.74 (0.65–6.48)
Oral cavity	12	0.94 (0.09)	0.97 (0.72–1.00)	12.67 (9.79)	11.56 (0.65–29.11)
Parotid L	12	0.97 (0.03)	0.97 (0.90–1.00)	8.96 (5.79)	9.44 (0.29–16.57)
Parotid R	12	0.96 (0.02)	0.96 (0.92–0.99)	10.33 (9.37)	7.50 (0.64–29.54)
Spinal cord	12	0.95 (0.07)	0.99 (0.78–0.99)	8.70 (16.12)	2.59 (0.29–47.74)
Submandibular gland L	11	0.93 (0.12)	0.97 (0.69–1)	5.75 (6.17)	3.92 (0.29–16.04)
Submandibular gland R	11	0.95 (0.05)	0.96 (0.85–0.99)	4.39 (1.69)	4.36 (2.55–7.44)
Thyroid	12	0.88 (0.11)	0.90 (0.69–0.99)	6.71 (3.49)	6.24 (2.55–11.21)

DSC: Dice similarity coefficient; HD: Hausdorff distance; SD: standard deviation; R: right; L: left.

## Data Availability

Data could be provided upon reasonable request from corresponding author.

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
