# Peer review of "Clinical Validation of a Deep-Learning Segmentation Software in Head and Neck: An Early Analysis in a Developing Radiation Oncology Center"

_ijerph, 2022, doi:10.3390/ijerph19159057_

Round 1

Reviewer 1 Report

First, I would like to thank the authors for allowing me to review their manuscript "Clinical validation of a deep-learning segmentation software in head and neck: an early analysis in a developing Radiation Oncology center".

In the study, a commercial software "Limbus AI" was investigated. The software can segment CT images automatically. This is particularly important for planning radiotherapy for head and neck tumors, according to the authors.

Major points:

- I miss the clinical implication. The accuracy of the segmentation is presented without somehow discussing in more detail whether it has an impact on therapy. The study design is inappropriate here. One would first have to have several subjects (at least 3) perform the segmentation and then compare the mean of the human against the machine. The authors apparently did not do this. The discssion is accordingly also only very superficial. Unfortunately.

- The study design described cannot really answer the question. There is a lack of subjects showing that inter-rater variability decreases due to AI. There is also nothing written about saving time. Why not? Have you ever measured the time advantage that this has brought?

- 12 cases are not enough. There should be significantly more so that a statistical evaluation is possible. The artificial inflation with 274 segmentations does not increase the number of cases per structure. I think there should be at least 30 CTs.

- Material and Methods section is very poorly described. Some important information is missing. How were the segmentations exported from Limbus AI to Varian Eclipse? As STL? As labelmap? Which software was used to calculate HD and DSC?

- Also when the authors refer to the guideline with "The contours generated by the AS were manually reviewed according to international consensus guidelines for H&N OARs contours [9,12,24,25]". How many people was this reviewed by? There should be at least two. In the introduction, the authors themselves write that there is inter-observer variability. This needs to be addressed and explained quite clearly. Otherwise, the results are just comparing a DL model with a single investigator.

Minor points:

- The authors write "Although their incidence is relatively low" referring to HNSCC. This is not true. It is the 6th most common tumor worldwide.

- The authors write "in some specific sub-sites of the head and neck region such as nasopharyngeal carcinoma." Yes, specifically HNSCC in the oropharynx. In the oral cavity, surgery is the treatment of choice. In the larynx only in higher stages.

- In Figure 1 the y-axis is not correct. Please correct it.

- In Figure 2 the y-axis is not correct either. You should indicate your mean 95% HD in mm.

- Figure 1 and 2 please also show the single values as points. All in all Figure 1 and 2 is very unreliable. Very many similar colors etc. I suggest to make 4 subfigures out of each with thematically similar contents. Brain/Brainstem/Globe_L/Globe_R/Chiasm/OPtic_Nerve could be a group for example.

Author Response

Point-by-point response to the comments of the reviewers

REVIEWER 1:

In the study, a commercial software "Limbus AI" was investigated. The software can segment CT images automatically. This is particularly important for planning radiotherapy for head and neck tumors, according to the authors.

A: We thank the reviewer for the appreciated comments.

Major points:

- I miss the clinical implication. The accuracy of the segmentation is presented without somehow discussing in more detail whether it has an impact on therapy. The study design is inappropriate here. One would first have to have several subjects (at least 3) perform the segmentation and then compare the mean of the human against the machine. The authors apparently did not do this. The discussion is accordingly also only very superficial. Unfortunately.

A: It is widely known in literature that manual segmentation can suffer of large inter-observer variability, so the variability in manual delineation among several subjects is out of the aim of this study. Our goal is to validate in the clinical context of a new Radiotherapy center the impact a new auto-contouring tool, able to make faster the clinical workload without comprising the quality of delineations. Such concepts have been integrated in the new version of the manuscript to make the reader aware of these aspects. (Page 2 – lines 82-88)

- The study design described cannot really answer the question. There is a lack of subjects showing that inter-rater variability decreases due to AI. There is also nothing written about saving time. Why not? Have you ever measured the time advantage that this has brought?

A: As already discussed in the previous comment, the inter-observer variability in manual delineation is a well-known aspect in Radiotherapy and one of the AI advantages is to increase the consistency between different observers: to support this, we included a new period in the introduction where some references stating this have been added. As regards time, we agree with the reviewer that the time saving is a crucial aspect, but having this study a retrospective nature, it was not possible to obtain this data. However, looking at literature, such aspect has been already investigated in a recent experience published on the Green Journal which has quantified the time saving in 26 minutes for Head and Neck by using this software. We thank the reviewer for having pointed out such aspect, that was included in the new discussion of the manuscript. (Page 8 – lines 254-259)

- 12 cases are not enough. There should be significantly more so that a statistical evaluation is possible. The artificial inflation with 274 segmentations does not increase the number of cases per structure. I think there should be at least 30 CTs. 

A: We thank the reviewer for the appreciated comments. This is a preliminary study based on the initial implementation of auto-contouring in a developing center including all patients actually treated. This aspect was also clearly stated as limitation in the discussion section. (Page 8 – lines 251-254)

- Material and Methods section is very poorly described. Some important information is missing. How were the segmentations exported from Limbus AI to Varian Eclipse? As STL? As labelmap? Which software was used to calculate HD and DSC? 

  1. The Material and Methods section has been integrated with the missing information. In particular the Limbus AI contours have been exported considering an RT Structure file in DICOM and transferred to Varian Eclipse. The MIM software (MAESTRO MIM) was used to calculate HD and DICE. Such informations were integrated in the new version of the manuscript. (Page 3/4 – lines 100-160)

- Also, when the authors refer to the guideline with "The contours generated by the AS were manually reviewed according to international consensus guidelines for H&N OARs contours [9,12,24,25]". How many people was this reviewed by? There should be at least two. In the introduction, the authors themselves write that there is inter-observer variability. This needs to be addressed and explained quite clearly. Otherwise, the results are just comparing a DL model with a single investigator.

A: We thank the reviewer for the appreciated comments, we modified the manuscript accordingly. (Page 1 – lines 28-29)

Minor points:

- The authors write "Although their incidence is relatively low" referring to HNSCC. This is not true. It is the 6th most common tumor worldwide.

A: We thank the reviewer for the appreciated comments, we modified the manuscript accordingly. (Page 2 – lines 51-53)

- The authors write "in some specific sub-sites of the head and neck region such as nasopharyngeal carcinoma." Yes, specifically HNSCC in the oropharynx. In the oral cavity, surgery is the treatment of choice. In the larynx only in higher stages. 

A: We thank the reviewer for the appreciated comments, we modified the manuscript accordingly. (Page 2 – lines 55-57)

- In Figure 1 the y-axis is not correct. Please correct it.

  1. Thanks, the y-axis in Figure 1 has been corrected.

- In Figure 2 the y-axis is not correct either. You should indicate your mean 95% HD in mm.

  1. Thanks, the y-axis in Figure 2 has been corrected.

- Figure 1 and 2 please also show the single values as points. All in all Figure 1 and 2 is very unreliable. Very many similar colors etc. I suggest to make 4 subfigures out of each with thematically similar contents. Brain/Brainstem/Globe_L/Globe_R/Chiasm/OPtic_Nerve could be a group for example.

We thank the reviewer for such request, all the figures were divided in subfigures as suggested

REVIEWER 2:

The paper presents a clinical validation study of deep-learning segmentation software in head and neck.

A: We thank the reviewer for the appreciated comments.

The following comments need to be addressed to improve the paper.

- The authors should state clearly the contribution of their research.

A: We thank the reviewer for the appreciated comments, we modified the manuscript accordingly. (Page 8/9 – lines 267-268)

- The name of the commercial software evaluated should be mentioned right from the beginning of the paper, in the abstract and introduction.

A: We thank the reviewer for the appreciated comments, we modified the manuscript accordingly. (Page 1 – lines 30-32)

- Section 2 is very short. The authors need to include a description of the theory of the methods used in this paper.  - Also, the datasets were not described in Section 2 properly (even the number of patients is not mentioned). The datasets should be described extensively in Chapter 2. The authors are recommended to have two subsections 2.1 to describe datasets, and 2.2 to describe the method used theoretically. - Also, the evaluation metrics should be explained in equations. I suggest including subsection 2.3 to describe the evaluation metrics.

A: We thank the reviewer for the appreciated comments, the methods has been divided in three subsections and detailed as request. (Page 3 – lines 100-161)

- Only two evaluation metrics are used. I suggest adding others such as confusion matrix, or TP, TN, FP, FN, sensitivity, specificity, etc. 

A: We thank the reviewer for the appreciated comments. The evaluation metrics suggested were used in the past but DICE and HD are more appropriate for contours evaluation: TP, TN, sensitivity and specificity are more in line with predictive models and ROC curves. A complete description has been reported in the new methods section. (Page 3 – lines 140-161)

- Writing must be improved. There are many paragraphs that consist of only one sentence (like the first and second paragraphs in the introduction, paragraph at line 98, and many others). Authors should fix this and all across the paper.

A: We thank the reviewer for the appreciated comments. We modified the manuscript accordingly.

- line 94: "valuated" should be evaluated

A: We thank the reviewer for the appreciated comments, we modified the manuscript accordingly.

- line 127: The HD is a quantification of the distance between two contour surfaces, The 95% HD ... --> the comma should be replaced by a period.

A: We thank the reviewer for the appreciated comments, we modified the manuscript accordingly.

- In the present manuscript the authors aimed to investigate the clinical impact of patients with primary resectable pancreatic cancer that underwent chemoradiotherapy.

Altogether this is a well-conducted retrospective analysis that could be considered for publication. However, some issues need to be addressed to improve the quality of the analysis, as commented below:

A: We thank the reviewer for the appreciated comments.

- Abstract and Conclusion: The conclusion needs to be rephrased to be more balanced, considering the current evidence on adjuvant chemotherapy trials. 

A: We thank the reviewer for the appreciated comments, we modified the manuscript accordingly. (Page 2 – line 71-73) Page 11 – line 291-298)

- Methods: The criteria for prescribing adjuvant CRT in this series should be clearly described

A: We thank the reviewer for the appreciated comments, we have described the prescribing criteria. (Page 5 – line 151/154)

- Introduction: A brief description of the range of median OS reported ín randomized studies using more aggressive chemotherapy (ESPAC4, PRODIGE, S1 trial) needs to be inserted for readers unfamiliar with the field.

A: We thank the reviewer for the appreciated comments. As requested we have widely described the OS rates. (Page 3 – line 99/112)

- Results: Please, briefly provide the range of R1 resection rates described in phase III trials to date. It should be commented that the definition of R1 resection varied among the trials.

A: We thank the reviewer for the appreciated comments, we added the requested information in the discussion. (Page 10 – line 261-271)

- Discussion: Similarly, to the above comment, a more critical discussion of the present findings compared to the literature data of adj chemotherapy trials needs to be performed.

A: We thank the reviewer for the appreciated comments, we added the requested considerations. (Page 10 – line 264-269)

- Discussion: the authors should describe the current guidelines and practice. What is the future of RT/CRT in this disease? Which trials with RT/CRT are currently conducted? Which patients could potentially benefit from adj CRT?

A: We thank the reviewer for the appreciated comments, we added the requested considerations. (Page 11 – line 279-286)

- Some minor typo errors need to be corrected. 

A: We thank the reviewer for the appreciated comments, we provided corrections.

Reviewer 2 Report

The paper presents a clinical validation study of deep-learning segmentation software in head and neck.

The following comments need to be addressed to improve the paper.

- The authors should state clearly the contribution of their research.

- The name of the commercial software evaluated should be mentioned right from the beginning of the paper, in the abstract and introduction.

- Section 2 is very short. The authors need to include a description of the theory of the methods used in this paper. 

- Also, the datasets were not described in Section 2 properly (even the number of patients is not mentioned). The datasets should be described extensively in Chapter 2. The authors are recommended to have two subsections 2.1 to describe datasets, and 2.2 to describe the method used theoretically.

- Also, the evaluation metrics should be explained in equations. I suggest including subsection 2.3 to describe the evaluation metrics.

- Only two evaluation metrics are used. I suggest adding others such as confusion matrix, or TP, TN, FP, FN, sensitivity, specificity, etc. 

- Writing must be improved. There are many paragraphs that consist of only one sentence (like the first and second paragraphs in the introduction, paragraph at line 98, and many others). Authors should fix this and all across the paper.

- line 94: "valuated" should be evaluated

- line 127: The HD is a quantification of the distance between two contour surfaces, The 95% HD ... --> the comma should be replaced by a period

Author Response

(The authors gave the same response as above.)

Round 2

Reviewer 2 Report

-The authors did not fix all comments as they mentioned in their responses.

For example, they were asked clearly to change the paragraphs with one statement such as the first paragraph in the introduction but that was not done. Even in the discussion there are many paragraphs with one statement..

The added statement in the conclusion is not about the contribution of this research (it is about the software they are using). Also, such statement should appear in the introduction not the conclusion.

Details about the patients (such as number of patients) was not mentioned in the Methods section, although it was clearly requested.

Author Response

Point-by-point response to the comments of the reviewers

REVIEWER 2:

The authors did not fix all comments as they mentioned in their responses. For example, they were asked clearly to change the paragraphs with one statement such as the first paragraph in the introduction but that was not done. Even in the discussion there are many paragraphs with one statement.

A: We thank the reviewer for the appreciated comments, we modified the manuscript accordingly. (Page 2 – Lines 50 – 53 and 74 – 77;Page 8 – Lines 191-193 and 198 – 202; Page 9 – Lines 224-239)

The added statement in the conclusion is not about the contribution of this research (it is about the software they are using). Also, such statement should appear in the introduction not the conclusion.

A: We thank the reviewer for the appreciated comments, we modified the manuscript accordingly. (Page 8 – Lines 260 – 263)

Details about the patients (such as number of patients) was not mentioned in the Methods section, although it was clearly requested.

A: We thank the reviewer for the appreciated comments. The number of patients was already declared in the results section. (Page 4 – lines 161-164)

Round 3

Reviewer 2 Report

Thanks to the author for revising the paper. However, it seems that they did not get my previous comment. The revised text is not better than the older one. Writing has to be improved (should write paragraphs not single statements here and there. Also, in many places, semi-colon is used instead of full stop). An English native speaker professional has to proofread the manuscript. It cannot be accepted it in its current shape.

The number of patients should be mentioned in the data section not the results section.

Equations should be numbered and written using the equation editor. For example, the equation in line 153 is not clear. What do the authors mean by max min? this is not clear. Authors should explain this.

line 177 about Figure 1? is this a mistake?

Author Response

Point-by-point response to the comments of the reviewers

REVIEWER 2:

Thanks to the author for revising the paper. However, it seems that they did not get my previous comment. The revised text is not better than the older one. Writing has to be improved (should write paragraphs no single statements here and there. Also, in many places, semi-colon is used instead of full stop). An English native speaker professional has to proofread the manuscript. It cannot be accepted it in its current shape.

A: We thank the reviewer for the appreciated comments, we modified the manuscript accordingly. The manuscript was handled by a English professional and substantial re-written as recommended by the reviewer. All the changes are highlighted in the new version of the manuscript. Most of the periods were rephrased and modified without altering the messages. We hope that the new version of the manuscript can be now considered valuable

The number of patients should be mentioned in the data section not the results section.

A: We modified the manuscript accordingly. (Page 3 – Line 103 and Page 4 – Line 161)

Equations should be numbered and written using the equation editor. For example, the equation in line 153 is not clear. What do the authors mean by max min? this is not clear. Authors should explain this.

A: We thank the reviewer for having pointed out this aspect. All the equations have been numbered. The part related to the Hausdorff distance has been totally rewritten and modified, also including a new formulation clearer to avoid possible source of misunderstanding.

The HD is a quantification of the maximum distance between the two contour surfaces (manual and automatic). Such indicator returns the maximum distance between two contours performing the calculation to the nearest point in both directions, from contour A to B and vice versa.

2                    

Where h(A,B) is the Euclidean distance of the voxels a and b belonging to the contours A and B, whose formulation is:

3                    

line 177 about Figure 1? is this a mistake?

A: We thank the reviewer for the appreciated comments, we removed the sentence. (Page 4 – Lines 176)
